# Access to essential psychotropic medicines in Addis Ababa: A cross-sectional study

Molla Teshager[1], Mesfin Araya[2], Teferi Gedif Fenta[1]*

1 Department of Pharmaceutics and Social Pharmacy, School of Pharmacy, College of Health Sciences, Addis Ababa University, Addis Ababa, Ethiopia, 2 Department of Psychiatry, School of Medicine, College of Health Sciences, Addis Ababa University, Addis Ababa, Ethiopia

* tgedif@gmail.com

## Abstract

### Background

Mental disorders are becoming a growing public health problem worldwide, especially in low- and middle-income countries. Regular and adequate supplies of appropriate, safe, and affordable medications are required to provide quality mental health services. However, significant proportions of the population with severe mental disorders are not getting access to treatment. Among others, the availability and affordability of psychotropic medicines are significant barriers for many patients in meeting their medication needs. This study aimed to assess the availability, prices, and affordability of essential psychotropic medicines in the private and public health sectors of Addis Ababa, the capital city of Ethiopia.

### Methods

A cross-sectional study design was used in 60 retail medicine outlets from the public and private sectors. Stratified random and quota sampling were applied to select the retail outlets. Data was entered and analyzed using the preprogrammed WHO/HAI workbook and SPSS V.25.

### Results

The mean availability of Lower Priced Generic (LPG) psychotropic medicines was 24.33% in Addis Ababa (28.7% in the public sector and 19.80% in the private sector). The Patient prices for the LPG ranged from 0.52–6.43 MPRs in public and 1.08–24.28 MPRs in private sectors. Standard treatment costs varied from 0.1–7.8 days' wages in public and 0.8–25 days' wages in private sectors for the lowest-paid government worker to purchase a month's supply.

### Conclusions

Essential psychotropic medicines were poorly available, with high prices and low affordability in Addis Ababa. An efficient supply across all levels of care and financial protection for essential medicines should be in place to ensure access.

**Data Availability Statement:** All relevant data are within the paper and its Supporting Information files.

**Funding:** The Authors received no specific funding for this work.

**Competing interests:** The authors declared that no competing interests exists.

**Abbreviations:** FMHACA, Food, Medicine and Healthcare Administration and Control Authority; HAI, Health Action International; IOM, Institute of Medicine; MOH, Ministry of Health; MSH, Management Sciences for Health; OECD, Organization for Economic Cooperation and Development; PFSA, Pharmaceutical Fund and Supply Agency; UN, United Nations; WHO, World Health Organization.

## Introduction

Mental health matters most for the well-being of every one of us. Moreover, mental health is a prerequisite for the well-being of individuals, societies, and the country at large [1, 2]. Despite these facts, many people worldwide are affected by severe mental disorders [3]. Mental illness is the leading cause of non-communicable disorders in Ethiopia. It accounts for 11% of the total burden of diseases in the country, and about 8000 people commit suicide every year [4, 5]. Also, evidence from cohort studies shows that persons with severe mental health conditions have died about 30 years earlier than the general population in Ethiopia [6]. Moreover, despite the increased demand for mental healthcare services, the treatment gap is as high as 90% [7], although there are medicines in the market for most mental illnesses [8, 9]. Several effective medicines are available for mental disorders, but not all "effective" drug therapies are essential [10].

Essential psychotropic medicines are those medicines that satisfy the priority mental health care needs of the population. These medicines should be made available at all levels of health care: continuously, in adequate amounts, in the appropriate dosage forms, with assured quality and adequate information, and at prices, individuals and the community can afford [11]. Access to essential psychotropic medicines is part of a fundamental human right [12, 13], and ensuring access to medicines for mental disorders can be beneficial not only for the patients themselves but also for employers through reduced absenteeism and higher productivity; for family members and friends, lowering the burden of care; and for government, through reduced social security benefits [14, 15]. However, compared to other essential medicines generally, the availability and affordability of medicines for mental disorders are even worse for the populations of low-income countries where mental illnesses cause enormous morbidity, disability, and mortality [1, 16, 17].

As a result, a lack of access to essential psychotropic medicines can significantly contribute to the public health burden [18]. The right to health, including access to medicine, is a fundamental human right recognized by numerous international human right laws [19]. Likewise, as one of the WHO member states, Ethiopia indicated 'health' as a fundamental human right in its constitution [20]. The national pharmaceutical policy has been in place since 1993 to ensure access to essential medicines as a major objective [21]. However, Ethiopia has no separate mental health policy [22].

The essential psychotropic medicine list is organized as per the pharmacotherapy classes of medicines that mimic the World Health Organization (WHO) model list. The list contains only generic medicines in alphabetical order, without identifying the healthcare facilities' levels where some specialty medicines can be used. At the national level, they are compiled as part of the general essential medicines list and have not been compiled as a standalone document [23].

In Ethiopia, in recent years the number of patients seeking care for a wide range of mental illnesses has grown and has also increased the need for psychotropic medications [4, 5, 24]. However, most of the resources (staff, budgets, and beds) for mental healthcare services are located in Addis Ababa City. The only mental hospital in the country that serves the whole nation as the highest referral and training center in mental health is also found in the city [4]. Besides, studies confirmed that the prevalence of common mental disorders in Addis Ababa is higher than the national prevalence rate [25, 26]. This might be attributed to the city's nature of urbanization and lifestyle changes [1, 27]. Having this in mind, ensuring access to basic psychotropic medicines has been considered essential for the mental healthcare services provided to patients in the city [1]. Accessibility of treatments for mental disorders is relatively low for numerous reasons despite the considerable burden of mental illness globally [8, 28].

Evidence showed that access to essential medicines could be mainly affected by stock-out, high prices, and unaffordability, amongst others [29]. Especially in the public sector, availability is relatively low. This, in turn, results in medicines being purchased with higher out-of-pocket expenses from private medicine outlets [4, 28]. Generally, about 90% of the population in low and middle-income countries rely on out-of-pocket expenses for their pharmaceutical needs due to inadequate public health services and lack of health insurance [30]. Similarly, due to an inadequate budget, centralization of mental healthcare services, and lack of human resources, medicines for mental illnesses are not continually available in Ethiopia. Hence, most mental health seekers remain under-treated [9, 31, 32], increasing the risk of relapse, re-hospitalization, comorbidities, and premature death [24].

At the local and national levels, few studies have examined essential medicines' availability, price, and affordability. In particular, studies examining the availability, prices, and affordability of psychotropic medicines are few [15, 32, 33]. As a result, policymakers find it challenging to set priorities and test interventions to improve access to mental disorders treatments [34]. Above all, further efforts to expand access to mental healthcare services can be fruitless without ensuring essential psychotropic medicines' sustainable availability and affordability [35]. Thus, this study aimed to measure essential psychotropic medicines' availability, price, and affordability at public and private medicine retail outlets in Addis Ababa.

## Methods

An institution-based cross-sectional study was carried out between 30[th] July and 18[th] September 2019 using WHO/HAI tools [30] to collect data on prices and availability from public and private sectors in Addis Ababa, Ethiopia's capital city. The IRBs of the School of Pharmacy and Addis Ababa City Health Bureau approved the study. Data collectors approached the heads/owners of the selected retail outlets, explained the purpose of the survey, and ensured that the identifier information for each participating institution would not be used in describing the result. Consequently, informed verbal consent was obtained from the participants.

Addis Ababa has a population of more than 3.8 million [36]. During the study period, there were 1919 health facilities in Addis Ababa, of which 110 were public healthcare facilities and 558 were private retail pharmacies. There are two levels of retail outlets in the Ethiopian healthcare system. Licensed diploma holders open drug shops with pharmacy training and are not allowed to store and sell psychotropic medicines. The term pharmacy, in this study, entails retail pharmacies that are permitted to store and dispense psychotropic medicines. On the other hand, in this study medicine retail outlets include public medicine dispensing units and private retail pharmacies.

The source population was 668 medicines retail outlets from which the sampled outlets had been selected using stratified, random sampling quota and purposive sampling. The sample size was determined based on the WHO/HAI manual [30]. Accordingly, 60 retail outlets, 30 each from the public and private sectors, were included (Table 1).

### Selection of the medicines to be surveyed

The psychotropic medicines included in the present survey were based on the 20[th] edition of the WHO Model List of Essential Medicines, the 5[th] National Essential Medicine Lists of Ethiopia, and the MSH-2015 price indicator [23, 37, 38]. Besides, expert opinion from Addis Ababa University using focus group discussion and feedback from Health Action International was obtained via electronic communications. Despite the variations in their dosage forms and strengths, 26 essential generic psychotropic medicines were included in Ethiopia's national medicine list [23]. Moreover, out of 26 essential psychotropic medicines, 11 were not

**Table 1. Distribution of health facilities in each sub-city of Addis Ababa, 2019.**

| Subgroups | All Sub Cities | Study Population | | Selected Sub-city | Sample Size | | Total |
|---|---|---|---|---|---|---|---|
| | | Public | Private | | Public sector | Private sector | |
| Group:1 | Bole | 10 | 65 | Bole | 10 | 14 | 24 |
| | AkakiKality | 10 | 38 | | | | |
| | Yeka | 15 | 71 | | | | |
| Group:2 | Kirkos | 10 | 30 | Kolfe | 10 | 8 | 18 |
| | Gullele | 13 | 130 | | | | |
| | Nefas Silk | 10 | 26 | | | | |
| | KolfeKeranyo | 11 | 38 | | | | |
| Group:3 | Lideta | 8 | 20 | Addis Ketema | 10 | 8 | 18 |
| | Arada | 13 | 102 | | | | |
| | Addis Ketema | 10 | 38 | | | | |
| **Total** | | **30** | **30** | | | | **60** |

Sources: Addis Ababa Health Bureau's FMHACA 2019.

registered by the medicine regulatory authority of Ethiopia. This might be due to local manufacturers' and importers' lack of interest in supplying these medicines [39].

Subsequently, all 26 psychotropic medicines with specific strength and dosage forms were used for the pilot test; six medicines were not found in any of the pilot retail outlets. Four of the six medicines are used only as emergency medicines at the inpatient level and are not stocked in most outpatient pharmacies. These medicines included: clonazepam 1 mg/ml injection, magnesium 50% in 20 ml injections, lorazepam 1 mg/ml injection, midazolam HCl 1 ml/ml injection. The other two, namely, bromazepam 3 mg and quetiapine 50 mg tablets, have not been found in all the retail outlets of the pilot study. Besides, bromazepam 3 mg and quetiapine prices were not found in the MSH-2015 price indicator. So, the six medicines were excluded from the survey list, leaving only 20 essential psychotropic medicines to be included in the study.

For each medicine in the survey, up to two products were considered: the originator brand (OB) and the lowest-priced generic equivalent (LPG). The OBs of psychotropic medicines were reviewed from previous studies [40–45]. Based on this evidence, the OBs of the medicines were selected-which had been given by the manufacturers of the respective products when they were marketed for the first time across the world. So the originator brand name used in this study does not stand for various brand names of a medicine that was given after the originator brand name patent right had expired. As a result, the majority of originator brands of medicines selected for this survey were not registered by the medicine regulatory authority of Ethiopia. This could be because the originator brands might have lost their exclusive market share after their equivalent generic medicines became available in the market [46]. In general, the therapeutic categories of medications used for this study were: anxiolytics, antipsychotics, antidepressants, antiepileptics, and mood stabilizers.

## Eligibility criteria of the medicine outlets

**Inclusion criteria.** Public health facilities that have outpatient pharmacies and licensed private retail pharmacies that are closer to public health facilities and expected to stock psychotropic medicines were included in the study.

**Exclusion criteria.** Public health facilities that only stock a small number of emergency psychotropic medicines; and private retail pharmacies that were located far from public health facilities were excluded from the study.

## Sampling strategy

The residents of Addis Ababa are economically heterogeneous. During this study, the city was administratively divided into ten sub-cities. The ten sub-cities are stratified into three sub-groups based on their relative per capita income and poverty status [47]. Accordingly, one representative sub-city from each sub-group was selected using a simple random sampling (S1 File).

In addition to economic status, accessibility to essential psychotropic medicines has also been affected by the type of health sector. The two main health sectors in Addis Ababa are the private and public. Thirty retail outlets from each sector were included in the study [30]. The specific retail outlets are selected using the proportional to size technique. Besides, the private outlets were selected based on their proximity to primary, secondary, and tertiary public health facilities surveyed (Table 1).

## Data collectors, procedures, and collection tool

Two trained pharmacists were employed for this study as data collectors. Data on the price and availability of psychotropic medicines was collected by physically inspecting the stock of the OBs and their LPG on the day of the survey using a modified form of the standardized WHO data collection tool [30]. For each medicine listed, information regarding the manufacturer, the pack size, and the pack price was recorded for both generics and innovator products (S2 File). The procurement price was gathered from the public-sector procurement agency. The prices were converted to US dollars using the buying exchange rate on 30th July/2019, the first day of data collection (1 USD = 29.0256 ETB).

## Data quality assurance

The medicine's price data collection form was pre-tested in 20% of the total sample size (n = 60) before the actual data collection. The medicines' list, strengths, and dosage forms were reviewed following the pre-test. Also, during the actual data collection, the questionnaire was revised for completeness of the forms and accuracy of unit prices each day. Double data entry and workbook auto checker were also employed to identify any discrepancies and to ensure the accuracy of the data entry process. Descriptive statistics were also used to clean the raw data.

## Data analysis

WHO/HAI price workbook-part-I and the Statistical Package for the Social Sciences (SPSS) version 25 were used to enter, edit, analyze and summarize the data. In order to measure the outcomes of this study, the following definitions were used. In this study, medicine availability was defined as the presence of the survey medicines at the specified strength and dosage form in selected retail outlets on the day of the survey. Availability was determined as the mean availability of individual medicines, group of medicines, product types (originator brand vs. generic), of medicines between sectors. The following ranges were used to describe percentage availability [48].

- < 30% = very low
- 30 –<50% = low
- 50–80% = fairly high
- >80% = high

In the price data analysis, median medicines' prices in local currency were used. MPRs could not be calculated until at least four retail patient prices were available from each sector for the medicine in question. The international price guide indicator of MSH has been used for this study to compare prices across countries to see the trend of change in medicine prices [38]. The median buyer unit price was used where no supplier prices were available. The median price ratio was obtained by dividing medicine's local median price by the international reference price converted to local currency by the equivalent buying exchange rate. i.e.

$$\text{Median Price Ratio (MPR)} = \frac{\textbf{Median local price}}{\substack{\textbf{International reference unit price} \\ \textbf{in local currency}}}$$

Thus, the ratio tells us how much greater or less the local medicine price is compared to the international reference price. However, there are no universally accepted interpretation of MPRs due to the difference in medicine price components and the different market systems in various countries. For this study, the following MPR cut-off points were used to indicate acceptable local price ratios based on the definition used in surveys performed in Ethiopia and elsewhere with similar methodology [49, 50].

- Public sector procurement price: MPR ≤ 1
- Public sector patient price: MPR ≤ 1.5
- Private retail pharmacy patient price: MPR ≤ 2

In Ethiopia, about 88% of patients buy medicines out of pocket from public pharmacies, which accounts for 47% of the households' out-of-pocket total expenditure [51, 52].

The affordability of standard treatments for five different mental health conditions was analyzed. The total monthly dose was determined by multiplying each medicine's commonly prescribed daily dose by 30 days. The affordability of treatments was assessed by considering the recently updated net salary of the lowest-paid government worker from the Federal Civil Service Authorities of Ethiopia, which came into effect on 8th July 2019, which was 973 birr/month, or 32.28$/month. Accordingly, the daily wage was determined by dividing the net salary for 30 days, ETB 32.43/day, or USD 1.12/day. The treatment that cost only 1-day income or less was deemed affordable [30].

## Results

### Availability of psychotropic medicines

The mean availability for 20 LPG psychotropic medicines in Addis Ababa city was 24.3%. Out of the 20 psychotropic medicines surveyed, only the originator brand of carbamazepine was found across the 60 outlets, and its mean availability was 2.4%. The overall mean availability of the basket of medicines across the sample of 30 public medicine outlets was 28.7% for the LPG and 2.8% for the OB. Among the 20 essential psychotropic medicines surveyed, one OB and 17 LPG medicines were found across the sampled public sector medicine retail outlets.

Medicines like alprazolam 0.5 mg tab, clozapine 25 mg tab, and lamotrigine 50 mg tablet were not found throughout the sampled public medicine retail outlets, and the availabilities for most of the surveyed medicines were below 50%. The percentage availability for the OB of carbamazepine was 56.70% which was much greater than the availability of its LPG (3.3%). Similarly, the finding showed that neither the LPG nor the OBs of 11 medicines was found in any private medicine retail outlets. However, the remaining nine medicines were found with availability ranging from 3.3% to 90.0% in medicine retail outlets in this sector. The overall

mean availability of psychotropic medicines in the private sector was 19.8% for the LPG and 2.0% for the OB medicines. The availability of Tegretol was 40%, but its LPG was 20%. Besides, as shown in Table 2, out of the 20 LPG medicines, only fluoxetine and sodium valproate were adequately available (> 80%).

Essential psychotropic medicines' availability was also analyzed. Accordingly, the mean availability of LPG medicines in health centers and hospitals was 24.2% and 51.0%, respectively. Similarly, the mean availability of OB was 2.4% and 5% for health centers and hospitals, respectively. Besides, about 85% of the surveyed LPG medicines were observed only in one specialized mental hospital, but the mean availability of LPG in the 29 non-mental healthcare facilities was only 26.7%. Despite the number of public health facilities found at various healthcare levels, the mean availability of OB medicines was meager. The availability of LPG medicines was improved as the level of care increased.

The availability of at least one essential psychotropic medicine from each therapeutic class was observed only in six public medicine retail outlets and none in the private sector. Most retail outlets in both sectors (n = 28/60) were stocking at least one medicine from the three classes alone (Fig 1).

## Price of psychotropic medicines

**Public procurement prices.** The procurement prices of one originator brand and 16 lowest-price generic medicines were obtained. Half of the LPG medicines (n = 8/16) were procured with a price < 1 MPR, whereas the others were procured with a price > 1 MPR. The MPRs for OB and LPG of carbamazepine 200 mg tab were 4.36 and 1.68 times the international reference price, respectively. Though 20 medicines were included in the survey, procurement price data was obtained only for 16 LPGs and one originator brand. The MPPR for 16 LPGs MPR was 0.96 (min 0.25 and max 4.83) (Table 3).

**Table 2. Availability of LPG medicines in public & private retail outlets (n = 60), Addis Ababa, 2019.**

| Medicines name, strength and dosage form | Public retail outlets (n = 30) | Private retail outlets (n = 30) |
|---|---|---|
| Alprazolam 0.5 mg tab | 0.0 | 0.0 |
| Amitriptyline 25 mg tab | 90.00% | 56.70% |
| Carbamazepine 200 mg tab | 3.30% | 20.00% |
| Chlorpromazine 100 mg tab | 76.70% | 13.30% |
| Clomipramine 25 mg cap | 3.30% | 0.0 |
| Clozapine 25 mg tab | 0.0 | 0.0 |
| Diazepam 5 mg tab | 80.00% | 0.0 |
| Fluoxetine 20 mg cap | 56.70% | 83.30% |
| Fluphenazine 25 mg/ml inj | 13.30% | 0.0 |
| Haloperidol 2 mg tab | 46.70% | 0.0 |
| Imipramine 25 mg tab | 33.30% | 0.0 |
| Lamotrigine 50 mg tab | 0.0 | 50.00% |
| Lithium CO3 300 mg cap | 6.70% | 0.0 |
| Olanzapine 5 mg tab | 20.00% | 3.30% |
| Phenobarbital 30 mg tab | 73.30% | 33.30% |
| Phenytoin 100 mg tab | 26.70% | 0.0 |
| Risperidone 1 mg tab | 16.70% | 46.70% |
| Sertraline 50 mg tab | 13.30% | 0.0 |
| Sodium Valproate 200 mg tab | 10.00% | 90.00% |
| Trifluoperazine 5 mg tab | 3.30% | 0.0 |

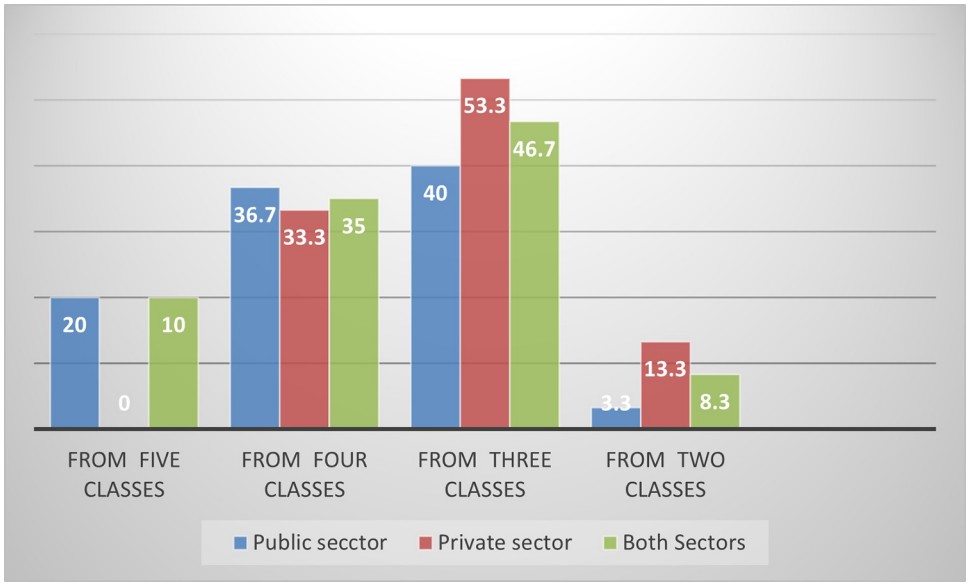

**Fig 1. Percent availability of at least one medicine from each therapeutic class in the retail outlets, Addis Ababa, 2019.**

## Retail prices of medicines in public and private sectors

The patient prices of the surveyed psychotropic medicines in both sectors are depicted in Table 4. MPR prices for 8 LPG medicines in public and 12 LPG medicines in private retail outlets were not determined. This is because either these medicines were found in less than four outlets, or they were not found in any of the sampled outlets. In the public sector, the median MPR amongst the 12 LPG medicines included in the price data analysis was 1.53. In contrast to the LPG, the MPR price for the OB of carbamazepine was 5.21. In the private sector, 8 LPG

**Table 3. Public sector procurement prices of medicines, Addis Ababa, 2019.**

| No. | Medicine Name | Medicine Type | Median Price Ratio | Median Price |
|---|---|---|---|---|
| 1 | Alprazolam 0.5 mg tab | LPG | 1.78 | 0.83 |
| 2 | Amitriptyline 25 mg tab | LPG | 1.94 | 0.47 |
| **3** | Carbamazepine 200 mg tab | **OB** | 4.36 | 2.34 |
| 4 | Carbamazepine 200 mg tab | LPG | 1.68 | 0.90 |
| 5 | Clomipramine 25 mg cap | LPG | 0.39 | 0.54 |
| 6 | Diazepam 5 mg tab | LPG | 0.25 | 0.07 |
| 7 | Fluoxetine 20 mg cap | LPG | 2.66 | 0.80 |
| 8 | Fluphenazine 25 mg/ml inj | LPG | 0.65 | 15.77 |
| 9 | Imipramine 25 mg tab | LPG | 0.73 | 0.38 |
| 10 | Lamotrigine 50 mg tab | LPG | 4.83 | 2.69 |
| 11 | Lithium $CO_3$ 300 mg cap | LPG | 1.03 | 0.85 |
| 12 | Olanzapine 5 mg tab | LPG | 0.26 | 0.72 |
| 13 | Phenobarbital 30 mg tab | LPG | 0.45 | 0.10 |
| 14 | Phenytoin 100 mg tab | LPG | 1.54 | 0.47 |
| 15 | Risperidone 1 mg tab | LPG | 3.08 | 3.35 |
| 16 | Sertraline 50 mg tab | LPG | 0.89 | 0.60 |
| 17 | Sodium Valproate 200 mg tab | LPG | 0.60 | 1.22 |

**Table 4. Prices of medicines in public and private retail outlets, Addis Ababa, 2019.**

| No. | Medicine Name | Medicine Type | Public (n = 30) | | Private (n = 30) | |
|---|---|---|---|---|---|---|
| | | | MORE | Median Price | MORE | Median Price |
| 1 | Amitriptyline 25 mg tab | LPG | 2.54 | 0.62 | 9.02 | 2.20 |
| 2 | Carbamazepine 200 mg tab | LPG | | | 4.89 | 2.63 |
| 3 | Carbamazepine 200 mg tab | **OB** | 5.21 | 2.8 | 11.17 | 6.00 |
| 4 | Chlorpromazine 100 mg tab | LPG | 0.6 | 0.25 | 1.08 | 0.45 |
| 5 | Diazepam 5 mg tab | LPG | 0.64 | 0.18 | | |
| 6 | Fluoxetine 20 mg cap | LPG | 3.51 | 1.05 | 10.37 | 3.10 |
| 7 | Fluphenazine 25 mg/ml inj | LPG | 1.43 | 34.73 | | |
| 8 | Haloperidol 2 mg tab | LPG | 1.64 | 1.04 | | |
| 9 | Imipramine 25 mg tab | LPG | 1.33 | 0.7 | | |
| 10 | Lamotrigine 50 mg tab | LPG | | | 24.28 | 13.53 |
| 11 | Olanzapine 5 mg tab | LPG | 0.52 | 1.42 | | |
| 12 | Phenobarbital 30 mg tab | LPG | 0.92 | 0.2 | 6.89 | 1.50 |
| 13 | Phenytoin 100 mg tab | LPG | 2.74 | 0.83 | | |
| 14 | Risperidone 1 mg tab | LPG | 6.43 | 7 | 4.64 | 5.05 |
| 15 | Sertraline 50 mg tab | LPG | 1.63 | 1.11 | | |
| 16 | Sodium Valproate 200mg tab | LPG | | | 3.97 | 8.00 |
| **Summary statistics** | | | **Public Sector** | | **Private Sector** | |
| Type of Medicines | | | LPG | OB | LPG | OB |
| Number of medicines included | | | 12 | 1 | 8 | 1 |
| Median MPR | | | 1.53 | 5.21 | 5.89 | 11.17 |
| 25%ile MPR | | | 0.85 | 5.21 | 4.47 | |
| 75%ile MPR | | | 2.59 | 5.21 | 9.36 | |
| Minimum MPR | | | 0.52 | 5.21 | 1.08 | |
| Maximum MPR | | | 6.43 | 5.21 | 24.28 | |

medicines were included to analyze patient prices, and the median MPR amongst the 8 LPG medicines was 5.89. The median MPR price for the OB of carbamazepine was 11.17. Like the public sector, only the OB of carbamazepine was also found in the private sector.

## Affordability of psychotropic medicines

Affordability was determined only for medicines which are available in at least four retail-outlets in each sector during the survey time. As indicated in Table 5, five LPG costs < 1 day's wage for treating the specified mental disorders in the public sector. However, the remaining LPG required more than one day's wage to buy the standard treatments in public health facilities. For example, treating epilepsy with the OB of carbamazepine was not affordable from both the public and private medicine retail sectors.

Similarly, some LPG treatment costs were surprisingly high when purchased in the private sector. For instance, treating epilepsy with lamotrigine 50 mg tablets required 25 days' wages, while treating bipolar disorders with sodium valproate costs 22.2 days' wages in private retail pharmacies. In the private sector, only the cost of chlorpromazine was affordable.

## Discussion

Ethiopia is a low-income country [53]. However, the prices of medicines have been guided by a free market system since the enactment of the first medicine policy in 1993 to ensure access to essential medicines through healthier competition [21, 46]. Therefore, medicine prices and

**Table 5. Affordability of standard treatments, Addis Ababa, 2019.**

| Mental Disorders | Medicines | Doses | Standard Treatments | | | Public Sector | | Private Sector | |
|---|---|---|---|---|---|---|---|---|---|
| | | | Treatment Duration (in Days) | Total doses per treatment | Product Type | Median Treatment Price | Days' Wages | Median Treatment Price | Days' Wages |
| Depression | Amitriptyline | 25 mg/tab | 30 | 90 | LPG | 55.80 | 1.7 | 198.00 | 6.1 |
| | Fluoxetine | 20 mg/cap | 30 | 30 | LPG | 31.50 | 1.0 | 93.00 | 2.9 |
| | Imipramine | 25 mg/tab | 30 | 60 | LPG | 42.00 | 1.3 | | |
| | Sertraline | 50 mg/tab | 30 | 30 | LPG | 33.18 | 1.0 | | |
| Epilepsy | Carbamazepine | | 30 | 90 | Brand | 252.00 | 7.8 | 540.00 | 16.7 |
| | | 200 mg/tab | | | LPG | | | 236.25 | 7.3 |
| | Lamotrigine | 50 mg/tab | 30 | 60 | LPG | | | 811.80 | 25.0 |
| | Phenobarbitone | 30 mg/tab | 30 | 90 | LPG | 18.00 | 0.6 | 135.00 | 4.2 |
| | Phenytoin | 100 mg/tab | 30 | 90 | LPG | 74.43 | 2.3 | | |
| Psychosis | Chlorpromazine | 100 mg/tab | 30 | 60 | LPG | 15.00 | 0.5 | 27.00 | 0.8 |
| | Fluphenazine | 25 mg/ml Inj | 30 | 1 | LPG | 34.73 | 1.1 | | |
| | Haloperidol | 2 mg/tab | 30 | 60 | LPG | 62.40 | 1.9 | | |
| | Olanzapine | 5 mg/tab | 30 | 90 | LPG | 127.58 | 3.9 | | |
| | Risperidone | 1 mg/tab | 30 | 30 | LPG | 210.00 | 6.5 | 151.50 | 4.7 |
| Anxiety | Diazepam | 5 mg/tab | 10 | 10 | LPG | 1.78 | 0.1 | | |
| Bipolar | Sodium valproate | 200 mg/tab | 30 | 90 | LPG | | | 720.00 | 22.2 |

mark-ups in the supply chain are left to the market competitors without control. Moreover, out-of-pocket expenditure for health care is the major source of financing in the country due to inadequate finance for public healthcare services and lack of health insurance mechanisms [51, 54, 55]. In 2011, community-based health insurance was introduced in Ethiopia, but its coverage is not wide enough to meet the aims of universal health coverage [55, 56]. The findings of this study concerning medicines' availability, prices, and affordability are discussed in the subsequent sections.

The findings of this study revealed that the overall mean availability of LPG psychotropic medicines in Addis Ababa was 24.3%. This value of mean percentage availability being below 30% indicates a very low availability of essential psychotropic medication for the study population [48]. Similarly, a national pharmaceutical sector assessment in 2016 demonstrated a very low availability of medicines for non-communicable diseases, including mental disorders [50]. This figure was far from the optimum availability indexes recommended by WHO which is above 80% [57]. This could conceivably be due to a lack of attention given to mental health problems and the availability of limited resources for improving access to essential psychotropic medicines [58, 59].

The mean availability of LPG psychotropic medicines in the public sector was 28.7%. This was generally very low but higher in hospitals (51%) than in health centers (24.2%). Comparing this result with the previous study, the median availability of medicines for chronic

illnesses, including mental disorders, in the hospitals (81.8%) was more than two times the median availability in the health centers (36.4%). In comparison, the median availability for other medicine groups, for instance, anti-infective medicines, was 87.5% in both health centers and hospitals [50]. This might imply an inequity of access to essential medicines for mental disorders at all healthcare levels. When comparing the percentage availability of individual medicines, lower availability was observed in this study than in the previous survey [60]. This implies that there is no improvement in the supply of access to essential psychotropic medicines while increasing demands for mental health medicines were evident [15].

Similarly, in the private sector, the mean availability of LPG psychotropic medicines was 19.8%, lower than what was found in the public sector covered in this study. This was also similar to the findings of a previous national study conducted in Ethiopia [50]. However, a study conducted in the private sector of Malawi, which included selected psychotropic medicines, reported that the availability of these medicines was higher than in the private retail outlets covered in this study. For instance, the availability of diazepam in Malawi's private sector was 100% [61], while in the present study, diazepam was not found in all private facilities of Addis Ababa. The highly controlled nature of the medications and the fact that they are less prescribed in the private sector might have contributed to their lower availability. Furthermore, the mean availability in the private sector in Addis Ababa was less than in the private sector of Saudi Arabia [62]. The overall mean availability in both sectors was far from the recommended minimum availability cut-off point [57].

With regard to the availability of the medicines from each therapeutic class, only 20% of the public retail outlets (n = 6/30) had at least one medicine from the five therapeutic groups of psychotropic medicines. Most of these public retail outlets that stock at least one medicine from each class were hospitals (n = 5/6). Most retail outlets were stocking at least one essential psychotropic medicine from the three classes alone from both sectors. This result was lower than the result found in the public sector of Mozambique, where 45.8% of the health facilities had at least one medicine from each class [34]. The minimum availability of at least one medicine from each therapeutic class, as suggested by WHO, was not achieved yet [63]. This limit is essential for providing minimum care to patients with mental health problems [64]. Besides, about 85% of the surveyed LPG medicines were available only in one specialized mental hospital. Similarly, a WHO study and a study done in Mozambique indicated that psychotropic medicines from each therapeutic category were more readily available in mental health facilities than in non-specialized healthcare facilities (26.7%) [17, 34], implying the non-decentralization of the mental healthcare service to all levels of healthcare in the country's health systems.

Regarding procurement prices, out of the 20 LPG medicines surveyed, the prices of 16 LPG medicines were found in the Public Supply Agency. Based on this procurement price data, the median MPR for the 16 LPG was 0.96 MPR, which was within the range of acceptable public procurement prices. However, when looking at the MPRs of each medicine, there were medicines purchased at ≥1MPR. While eight medicines were procured below 1 MPR (ranging from 0.25–0.89), the others were procured above 1 MPR (ranging from 1.03–4.83) compared with the IRPs [34]. This data suggests that the public procurement agency showed efficient procurement in 50% of the LPG medicines for which prices were obtained. When comparing the procurement prices of selected medicines (carbamazepine and phenytoin) with the previous study conducted in 2004, the MPRs of carbamazepine and phenytoin increased by 184.75% and 258.14%, respectively, from the MPRs of carbamazepine = 0.59 and phenytoin = 0.43 [60]. For some of the medications indicated, the purchasing efficiency of the public procurement sector in 2004 was better than in 2019. The need to explore the underlying factors that cause such expensive medicine procurement prices is imperative. Maintaining the public

procurement prices within the acceptable range (≤1MPR) is very crucial, particularly for low-income countries like Ethiopia, to improve access to essential medicines for mental disorders [48].

Concerning patient prices, the median patient price for 12 LPG out of the 20 surveyed psychotropic medicines in the public sector retail outlets was found to be 1.53 MPR. This price was slightly above the acceptable MPR price cut-off point used in this study (≤1.5 MPR), but it generally showed an improvement in the public sector retail outlets. However, for 6 LPG, the public patient prices were above 1.5 MPR (ranging from 1.63–6.43 MPRs). These high prices resulted from the same medicines purchased with costs above 1MPR during the procurement, except the for sertraline 50 mg tablet, which was expensive due to the excessive markup (83.15%). This data substantiated that procurement prices directly affect the patient prices of medicines in public health facilities and the mark-ups significantly affect the patient price [29]. The median MPR price of the OB for carbamazepine in the public sector was 5.21 MPR. However, the markup difference between the procurement price and the patient price of the OB was 19.7% which is below the average markup of the study area (22.04%) [50]. This indicates that the government applies a regressive markup pricing strategy (using lower markup for higher-priced products rather than fixed percentage markups) in public retail outlets [65].

Alternatively, the median MPR of patient prices in the private sector across 8 LPG medicines was 5.89 MPR which was more than the acceptable cut-off price used in this study and higher than the four MPRs of the IRPs [57]. This median MPR price showed 284.97% higher prices in the private sector than the median MPR of the patient price in the public retail outlets (1.53 MPR). Again, the MPR prices for 7LPG out of 8 LPG (87.5%) ranged from 3.97 MPR—24.28 MPR, which were > 2 MPR. So, it was only the MPR price of chlorpromazine 100 mg tablet (1.08 MPR) that was found within the acceptable price range for private retail outlets (≤ 2 MPR). Consequently, the low availability of medicines at public medicine retail outlets could directly impact access to essential psychotropic medicines as patients will be forced to buy these medicines from private pharmacies. The better availability in the public sector would pressure the private sector to lower the price of these generic medicines due to competition [46, 66]. In addition, the higher markup prices in the private sector might be due to lack of medicine price regulating mechanisms [21] and the variation in the procurement methods used between the two sectors [67, 68]. Similarly, the MPRs for amitriptyline and fluoxetine were increased by 215.38% and 625.17%, respectively, compared to the previous study in the private sector [50].

When looking at affordability, most of the psychotropic medicines found in both sectors were less affordable. However, 5 of 12 LPG medicines were found to be affordable in public retail outlets meeting the affordability targets of WHO [30]. The other 7 LPG were unaffordable, requiring 1.1 to 6.5 days' wages to cover a month of treatment. This shows that more than 58% of LPG found in public outlets were unaffordable for the lowest-paid unskilled government workers in Addis Ababa. So patients who could not afford such out-of-pocket costs would forgo the treatment, increasing the burden of mental illness [15, 69]. Higher medicine prices and low income of the lowest-paid workers contributed to the unaffordability of medicines for mental illnesses. The need to use one of the feasible price-reducing interventions on generic medicines in the public sector retails has been considered the only way to improve access for many low-income patients [65].

In the private outlets, only chlorpromazine 100 mg tablet was affordable out of the 8 LPG medicines in this sector. The present study showed that 87.5% of the existing treatments for mental disorders in private retails were non-affordable to many patients with mental health conditions in Addis Ababa [47]. Equally, the cost of medicines in private retail outlets documented in this study required more days' wages than in previous studies done in different

countries [60, 70]. The prices of critical medicines, including essential psychotropic ones, should not be left solely to market forces [10]. Otherwise, in a population with low-income levels and high medicine prices, it would mean that access to medicines is only affordable to a wealthy segment of the population [71].

Moreover, the costs of the OB were unaffordable in both sectors. In this regard, 16.7 days wage and 7.8 days wage are required to buy 30 days' supply from private and public retail outlets, respectively. This data was similar to a study done by WHO expert groups [71]. However, as the affordability data in this study was estimated from one OB alone, this report might be less conclusive about the affordability of OBs for psychotropic medicines. Moreover, even treatments that seem affordable in this report are too costly, at least for 18% of the Addis Ababa population earning less than the reference group [47].

Finally, this study provides an important picture of access to psychotropic medications. Stakeholders were involved during the proposal development, and their suggestions were included in the study. The clinical importance of the surveyed medicines has also been triangulated between the national essential list, the national standard treatment guideline, MSH 2015 price lists, and WHO essential lists. Moreover, variations in the type of health sector and the economic status of the general population for access to medicines in the study area were considered. However, this study has not been without limitations. The data on the availability of medicines was collected at a specific time. Thus, it does not reflect the average monthly or yearly availability of essential psychotropic medicines at the individual pharmacy outlets. Besides, medicines such as carbamazepine, phenobarbitone, phenytoin, risperidone, and sodium valproate, were found in different strengths and formulation types other than what was specified in the medicine price data collection form. As a result, such medicines available in different dosage forms and strengths were excluded from this study. Therefore, the non-availability and lower availability of these medicines may not make sense because they are available but with different strengths and dosage forms. Lastly, when other costs were also considered, treatments that appear relatively affordable in this study might overestimate affordability.

## Conclusion

The mean availability of LPG psychotropic medicines was generally very low in Addis Ababa–yet it was better in the public health facilities than in private retail outlets. Availability was lower in health centers than in hospitals. Most retail outlets were stocking at least one medicine from three classes alone in both sectors. Only the OB of carbamazepine was found in all medicine outlets with extremely low availability. The public procurement prices for half of the surveyed psychotropic medicines were high. Moreover, in both sectors, the patient prices for more medicines were high, and the costs for most of the standard mental treatments were considered unaffordable not only to the lowest-paid government workers but also to most populations in Addis Ababa living under the poverty line, especially in the private outlets.

## Supporting information

**S1 File. Grouping of sub-cities in Addis Ababa based on income and poverty status.**
(DOCX)

**S2 File. The customized WHO medicine price data collection form.**
(DOCX)

**S3 File. Datasets.**
(XLS)

## Acknowledgments

The authors would like to acknowledge the heads of all institutions that provided permission and study participants for giving information and making relevant documents available for review.

## Author Contributions

**Conceptualization:** Teferi Gedif Fenta.

**Data curation:** Molla Teshager.

**Formal analysis:** Molla Teshager.

**Methodology:** Molla Teshager, Mesfin Araya, Teferi Gedif Fenta.

**Supervision:** Mesfin Araya, Teferi Gedif Fenta.

**Validation:** Mesfin Araya.

**Writing – original draft:** Molla Teshager.

**Writing – review & editing:** Mesfin Araya, Teferi Gedif Fenta.

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
