## [Decision Letter · Decision Letter 0]

1 Sep 2022

PONE-D-22-17182Access to Essential Psychotropic Medicines in Addis Ababa: A Cross-Sectional StudyPLOS ONE

Dear Dr. Fenta,

Thank you for submitting your manuscript to PLOS ONE. After careful consideration, we feel that it has merit but does not fully meet PLOS ONE’s publication criteria as it currently stands. Therefore, we invite you to submit a revised version of the manuscript that addresses the points raised during the review process. The article has a correct structure, even the methodology used is suitable. I suggest you comply with the reviewers' comments; in addition, I suggest an English editing.

We look forward to receiving your revised manuscript.

Kind regards,

Andrea Cioffi

Academic Editor

PLOS ONE

Journal Requirements:

2. In the ethics statement in the Methods, you have specified that verbal consent was obtained. Please provide additional details regarding how this consent was documented and witnessed, and state whether this was approved by the IRB.

5. We note you have included a table to which you do not refer in the text of your manuscript. Please ensure that you refer to Table 1 in your text; if accepted, production will need this reference to link the reader to the Table.

Reviewers' comments:

Reviewer's Responses to Questions

**Comments to the Author**

1. Is the manuscript technically sound, and do the data support the conclusions?

Reviewer #1: Yes

Reviewer #2: Yes

2. Has the statistical analysis been performed appropriately and rigorously? 

Reviewer #1: Yes

Reviewer #2: Yes

3. Have the authors made all data underlying the findings in their manuscript fully available?

Reviewer #1: No

Reviewer #2: No

4. Is the manuscript presented in an intelligible fashion and written in standard English?

Reviewer #1: No

Reviewer #2: No

5. Review Comments to the Author

Reviewer #1: Dear authors, your research topic is very interesting and relevant. You have selected an appropriate methodology, but the methods section was not clearly described. Results presentation needs to be improved and made smarter maybe using figures. You have to avail the Excel worksheet in which you have entered data. You should also prepare supplementary files showing more details on prices and availability data.

Reviewer #2: Commendable efforts have been laid down in this valuable research. With minor though multiple edits as mentioned in the comments in the attached file, the manuscript can be made into a more clear and useful document for access research. I hope the suggestions made will be looked into positively. Looking forward to seeing your revised manuscript.

6. PLOS authors have the option to publish the peer review history of their article (what does this mean?). If published, this will include your full peer review and any attached files.

Reviewer #1: **Yes: **Thomas Bizimana

Reviewer #2: No

---

## [Author Response · Author response to Decision Letter 0]

24 Dec 2022

Access to Essential Psychotropic Medicines in Addis Ababa: A Cross-Sectional Study 

This Article is recommended for publication after improvement of the methods and results presentation sections. The study is scientifically relevant, the methodology employed is appropriate but not clearly described, and the way results are presented needs to be improved.

A. Major comments

I. General comments:

1. Carefully read the “Instructions for authors” concerning the format and how to cite references in the text. Ideally, use an automatic referencing system (e.g. Mendeley) and select the journal’s style. 

Response: EndNote referencing software with Vancouver referencing style was employed.

II. References: 

1. For all non-journal article references, provide the URL link and the date you lastly accessed it, to help reviewers check the correctness of the information.

Response: Accommodated.

III. Methods:

1. “1919 health facilities in Addis Ababa; of which 110 of them were public healthcare facilities and 558 were private retail pharmacies.” How about the other 1251 health facilities?

Response: Out of the total 1919 health facilities, the 1251 health facilities were private clinics or hospitals, NGO clinics or hospitals and other retail pharmacies, which were not included in the sample as they were not within the scope of the study.

2. “The study population was 668 medicines retail outlets”. How did you move from 1919 to 668 health facilities?

Response: The focus of the study was 110 public facilities’ dispensaries plus 558 private retail outlets making the study sample to be 668.

3. “The sample size was determined based on the WHO/HAI manual (29). Accordingly, 60 retail outlets; 30 each from the public and private sectors, were included.” Please, describe your methods clearly to help the readers understand what and how you have done. How did you come up with 60 from 1919 health facilities? 

Response: It is accommodated as suggested by the reviewers.

4. “Inclusion Criteria: Public health facilities that have outpatient pharmacies or dispensaries and private sector licensed retail pharmacies (closer to public health facilities) that are expected to stock psychotropic medicines were included in the study.

Exclusion Criteria: Public Health facilities that only stock a small number of emergency psychotropic medicine; and pharmacies in private clinics and hospitals or health facilities operated by private companies, such as mining companies, were excluded. Furthermore, drug stores were excluded from the study.” Please, consult your teachers of research methodology to understand the meaning of exclusion criteria: you can’t take beans A and B from a basket, while beans A and B have not been in the basket! Correctly write this section.

Response: accommodated.

5. “Thirty retail outlets for each sector were taken as an optimal sample size as per WHO’s recommendation.” Please, provide a reference with the URL link for this statement! 

Response: accommodated.

6. Table 2: why did you sample more health facilities from Bole sub-city more than from Kolfe and Addis Ketema (24/60)?

Response: sample allocation was done based on proportionate to size technique. In private sector, Bole has the higher number of private medicine retail outlets (65) than (38) and Addis Ketema (38).

B. Minor comments

I. Entire manuscript:

1. The text lines should be numbered to ease the review with reference to the line numbers.

Response: Accommodated 

II. Title page:

1. Author 1 and 2 have the same affiliation, use the same number “1”, then number 3 becomes number 2.

Response: Accommodated 

III. Abstract: 

1. Under the methods section, point out the types of health sectors surveyed.

Response: Accommodated

IV. Methods:

1. Rephrase the statement “An institution-based cross-sectional study was carried out between 30 July and 18 September 2019 using WHO/HAI tools to collect price and availability from public and private sectors in Addis Ababa, Ethiopia's capital city.” You have not collected prices and availability. Instead, you have collected data on prices and availability.

Response: Accommodated

2. Keep 1 table

Response: Accommodated

V. Results:

1. “As shown in Figure 1, the availability of at least one essential psychotropic medicine from each therapeutic class was only observed only in six public medicine retail outlets, …”. Delete the repeated word “only”.

Response: Accommodated

2. Some tables can be better replaced by figures.

Response: We have been tried to replace tables by figures as suggested. However figures were not suitable for some because of large number of variables. So we prefer to keep the tables as it is.

---

## [Decision Letter · Decision Letter 1]

1 Feb 2023

PONE-D-22-17182R1Access to Essential Psychotropic Medicines in Addis Ababa: A Cross-Sectional StudyPLOS ONE

Dear Dr. Fenta,

Thank you for submitting your manuscript to PLOS ONE. After careful consideration, we feel that it has merit but does not fully meet PLOS ONE’s publication criteria as it currently stands. Therefore, we invite you to submit a revised version of the manuscript that addresses the points raised during the review process. The article still needs some minor revisions.

We look forward to receiving your revised manuscript.

Kind regards,

Andrea Cioffi

Academic Editor

PLOS ONE

Journal Requirements:

Reviewers' comments:

Reviewer's Responses to Questions

**Comments to the Author**

1. If the authors have adequately addressed your comments raised in a previous round of review and you feel that this manuscript is now acceptable for publication, you may indicate that here to bypass the “Comments to the Author” section, enter your conflict of interest statement in the “Confidential to Editor” section, and submit your "Accept" recommendation.

Reviewer #1: All comments have been addressed

2. Is the manuscript technically sound, and do the data support the conclusions?

Reviewer #1: Yes

3. Has the statistical analysis been performed appropriately and rigorously? 

Reviewer #1: I Don't Know

4. Have the authors made all data underlying the findings in their manuscript fully available?

Reviewer #1: Yes

5. Is the manuscript presented in an intelligible fashion and written in standard English?

Reviewer #1: Yes

6. Review Comments to the Author

Reviewer #1: Few and minor edits before publication:

1. Cover page: Edit it according to the journal's template. Some information is not required.

2. Edit line 228 correctly.

3. Table 2 and 3: Remove medicines for which there are no data and describe them in the text. Delete "Table canted" and just make a summary. Data on prices should be provided for available medicines, otherwise they are meaningless. It is not clear why some medicines were available but no prices were recorded (e.g. chlorpromazine 100mg, haloperidol 2mg, ...).

4. Table 4 is misplaced.

5. Table 5: what is the meaning of absence of results on affordability? Provide an explanation in the text introducing this table.

6. Figure 1: No caption was inserted in the text. Results on availability should be expressed in terms of percentage.

7. References: URL missing for references No 2, 5, 20, 22, 23, 24, 46, 50, 51, and 55. A wrong URL was used for reference No 4, just copied from reference No 3. The URL for reference No 36 is not active.

7. PLOS authors have the option to publish the peer review history of their article (what does this mean?). If published, this will include your full peer review and any attached files.

Reviewer #1: **Yes: **Thomas Bizimana

---

## [Author Response · Author response to Decision Letter 1]

4 Mar 2023

Point by point response to reviewers’ comments

We are grateful to the reviewers for their constructive comments.

Reviewer #1: Few and minor edits before publication

1. Cover page: Edit it according to the journal's template. Some information is not required.

 Response: comment accommodated.

2. Edit line 228 correctly.

Response: accommodated

3. Table 2 and 3: Remove medicines for which there are no data and describe them in the text. Delete "Table contd." and just make a summary. Data on prices should be provided for available medicines, otherwise they are meaningless. It is not clear why some medicines were available but no prices were recorded (e.g. chlorpromazine 100mg, haloperidol 2mg,).

Response: Table 3 is reformatted as suggested but we prefer to retain Table 2 as it indicates availability of the selected medicines.

4. Table 4 is misplaced.

Response: accommodated

5. Table 5: what is the meaning of absence of results on affordability? Provide an explanation in the text introducing this table.

Response: Affordability was determined only for medicines which are available in at least four retail-outlets in each sector during the survey time and price data is obtained from these outlets to have ameangful calculation of MPR. An explanation is included in the revised version.

6. Figure 1: No caption was inserted in the text. Results on availability should be expressed in terms of percentage.

Response: Modified according to the suggestion

7. References: URL missing for references No 2, 5, 20, 22, 23, 24, 46, 50, 51, and 55. A wrong URL was used for reference No 4, just copied from reference No 3. The URL for reference No 36 is not active.

Response: Reference carefully looked into and the following changes are made in the revised version:

• No 2,5,20,22,23,24,46,50 and 55: URL included

• No 51 new reference replaced

• No 4 corrected

• R36 is still active

---

## [Editor Report · Decision Letter 2]

7 Mar 2023

Access to Essential Psychotropic Medicines in Addis Ababa: A Cross-Sectional Study

PONE-D-22-17182R2

Dear Dr. Fenta,

We’re pleased to inform you that your manuscript has been judged scientifically suitable for publication and will be formally accepted for publication once it meets all outstanding technical requirements.

Kind regards,

Andrea Cioffi

Academic Editor

PLOS ONE
---

## [Editor Report · Acceptance letter]

22 Mar 2023

PONE-D-22-17182R2 

Access to Essential Psychotropic Medicines in Addis Ababa: A Cross-Sectional Study. 

Dear Dr. Fenta:

I'm pleased to inform you that your manuscript has been deemed suitable for publication in PLOS ONE. Congratulations! Your manuscript is now with our production department. 

Kind regards, 

on behalf of

Dr. Andrea Cioffi 

Academic Editor

PLOS ONE